# Temperature-Dominated Driving Mechanisms of the Plant Diversity in Temperate Forests, Northeast China

**Yue Gu** [1,2] **, Shijie Han** [3,]***, Junhui Zhang** [1,]***, Zhijie Chen** [3]**, Wenjie Wang** [4]**, Yue Feng** [1]**, Yangao Jiang** [5] **and Shicong Geng** [1]

[1] Key Laboratory of Forest Ecology and Management, Institute of Applied Ecology, Chinese Academy of Sciences, Shenyang 110016, China; guyue740147940@126.com (Y.G.); lunamoon77@163.com (Y.F.); shiconggeng@163.com (S.G.)

[2] College of Resources and Environment, University of Chinese Academy of Sciences, Beijing 100049, China

[3] School of Life Sciences, Henan University, Kaifeng 475004, China; chan.chijay@gmail.com

[4] Northeast Institute of Geography and agroecology, Chinese Academy of Science, Changchun 130102, China; wjwang225@hotmail.com

[5] Experimental Teaching Center, Shenyang Normal University, Shenyang 110011, China; jiangyangao-jyg@163.com

*   Correspondence: hansj@iae.ac.cn (S.H.); jhzhang@iae.ac.cn (J.Z.); Tel.: +86-024-83970343 (J.Z.)

**Abstract:** Climate, topography, and tree structure have different effects on plant diversity that vary with spatial scale. In this study, we assessed the contribution of these drivers and how they affect the vascular plant richness of different functional groups in a temperate forest ecosystem in Northeast China. We investigated about 0.986 million plants from 3160 sites to quantify the impact of annual mean temperature, sunshine duration, annual precipitation, standard deviation of diameter at breast height, and forest type on richness of vascular plants (total species, tree, treelet, shrub, and herb, separately) using the gradient boosting model. The results show that annual mean temperature had the strongest impact on plant richness. The tree richness peaked at intermediate annual mean temperature and sunshine duration and increased with annual precipitation. The Shannon diversity index and Simpson dominance index increased with annual precipitation and standard deviation of diameter at breast height, decreased with sunshine duration, and peaked at intermediate annual mean temperature and forest type. The total richness and understory richness increased with annual mean temperature and standard deviation of diameter at breast height and peaked at intermediate sunshine duration and annual precipitation. A comprehensive mechanism was found to regulate the plant diversity in forest ecosystems. The relationship between tree richness and annual mean temperature with latitudinal effect could be affected by the differences in number and size of tree individuals, indicating that plant diversity varies with the utilization of energy. The force driving plant richness varied with the functional group due to the different environmental resource requirements and the life history strategies of plants layers.

**Keywords:** biodiversity; plant richness; climate; topography; tree structure; boosted regression trees; temperate forest; Shannon diversity index; Simpson dominance

---

## 1. Introduction

One of the earliest and most fundamental topics in both biology and ecology is the determination of species diversity [1–5], due to its importance for biodiversity conservation and the sustainable use and planning of nature reserves and forest management [6]. Species richness, as an important component of biodiversity, provides preliminary insights into the distribution of diversity and its crucial

drivers on different scales [7–9]. Many studies have explored the correlation between diversity and environmental features on species turnover, coexistence, and ecological succession [10–19]. However, the process by which the species composition and forest diversity varying with factors continuously remain ambiguous when considering large spatial scales [20].

Climatic gradient is considered as the main abiotic factor controlling large-scale species diversity [21–24]. Strong positive relationships between climate and plant richness are mainly impacted by temperature variations as well as latitude effects [23,25,26]. Other studies have suggested that cold temperatures may possibly be responsible for maintenance of the high species richness observed in some areas [27,28]; e.g., a greater plant diversity has been observed at cooler sites compared to warmer sites in near region [29–31]. This indicates that other factors impact the latitude effect at the regional scale. The influence of sunshine duration and sunlight intensity on primary productivity and productivity on diversity have also been widely recognized [32–37], whereas the relative importance of sunshine duration on diversity remains unclear.

Topographic factors have key effects on local species richness [38,39]. Slope affects diversity by influencing sunlight, soil fertility, and soil moisture [40,41]. Hump-shaped relationships were found in numerous taxa along altitudinal gradients [42–46]. Cases like this, in which richness peaks at medium gradients [47], are consistent with the intermediate disturbance hypothesis (IDH) and the niche-assembly hypothesis [48,49] for trees, shrubs, and herbs [47,50]. However, similar studies are common in the plant diversity of grasslands [51–53], but little is known about how topographic factors affect plant richness and whether other drivers affect hump-type diversity of forest on a regional scale.

Different local site conditions can satisfy the hydrothermal needs of diverse plant species and different richness patterns and environmental heterogeneity can be observed in the various taxa [23,31,54,55]. Plants in different functional groups (tree, shrub, herb, and treelet) have different environmental requirements when considering the shading effect of canopy [56,57]. These factors not only affect the structure and composition of the ecosystem, both directly and indirectly [58–63], but also the pattern of biodiversity over both time and space [64–68]. On a macroscopic scale, the variation in species (genera or families) richness reflects overlapping differences in the distributional range, whereas the edge of distribution limits the capacity of individuals to tolerate and occupy an environment [69]. This indicates how individuals develop and the kind of taxa they formed could also impact diversity [31,70–73]. However, reports examining the relative contribution of forest structure to plant diversity of tree, shrub, herb, and treelet are rare, especially considering large spatial scales.

The mountain forest in Northeast China, as one of the most species-rich temperate ecosystems, are influenced by natural factors over the last few decades, although part of forests surveyed had been under historical impact of forest management eighty years ago. All the sample plots we investigated are unmanaged and far away from the road. There are no other communication routes in the surrounding of the forests studied and the interference of urbanization, agriculture, and industry to the plots are also very small. Therefore, the forest is a typical case that can be used to characterize the impact of environmental heterogeneity on the level of biodiversity. The objectives of this study are to quantify and compare the contribution of climate, topography, and taxa structure to plant diversity using field survey data. We hypothesized that (I) plant richness increases with annual mean temperature and annual precipitation, but has an upper limit; (II) plant richness varies with sunshine duration in hump-shaped relationships; (III) the greater the standard deviation of diameter at breast height (DBH), the higher the plant richness; (IV) forest type can impact plant richness, but not very much relative to other factors; (V) topographic factors influence plant richness a little on large scale; and (VI) the relative contribution of the drivers to plant diversity and the responses vary with functional groups on a regional scale. This generated information can provide support for sustainable forest management and plant diversity conservation at regional scales [6], such as which forest types or community structures can be constructed to maintain optimal biodiversity in different environmental context.

## 2. Materials and Methods

### 2.1. Study Area

This study was conducted in temperate mountain forests, Northeast China (39°54′5″–53°19′12″ N and 117°11′3″–133°52′20″ W, Figure 1). The main standing trees are deciduous broadleaved trees (*Quercus mongolica*, *Tilia amurensis*, *Acer mono* and *Betula platyphylla* et al.), deciduous coniferous trees (*Larix gmelinii* and *Larix olgensis*), and evergreen coniferous trees (*Pinus koraiensis*, *Abies nephrolepis*). The main shrubs, for instance, are: *Rhododendron aureum*, *Lonicera japonica*, *Lespedeza bicolor*, *Euonymus alatus* et al. The main herbs, in turn, are *Arisaema heterophyllum*, *Saussurea japonica*, *Brachybotrys paridiformis*, *Oxalis corniculate* et al. The region is mountain monsoon forest of temperate continental climate zone and characterized by four distinctive seasons, with a warm-rainy summer and a cold-dry winter. The annual mean temperature varies by more than 10 °C with 9.2 °C in the south and −0.9 °C in the north. The annual precipitation decreases from about 1000 mm in the southeast to less than 400 mm in the northwest. Most of the rainfall is concentrated between June and September. The elevation ranges from 30 to 2073 m.

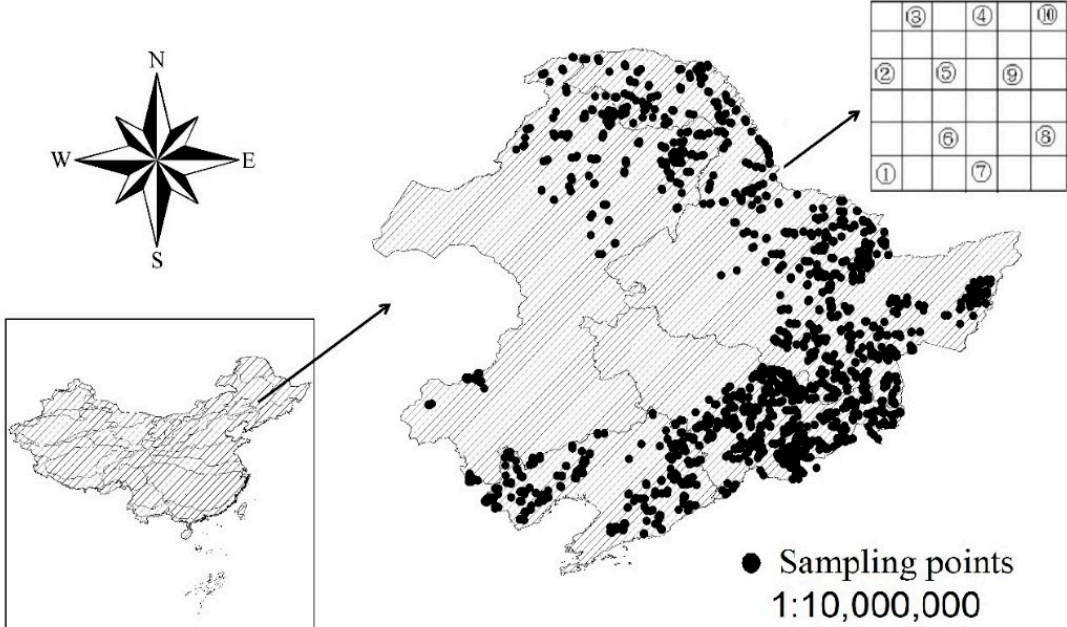

**Figure 1.** Sampling points in the study area.

### 2.2. Data Investigation

Our data were supplied from two special investigations of vegetation in summer (July–August) between 2008 and 2017. In total, 3160 standardized, 900 m² (30 m × 30 m) plots were randomly sampled in the mountain forests in Northeast China (Figure 1). In each plot, latitude, longitude, elevation, slope, and aspect were recorded before plant investigation. The height and the diameter at breast height (DBH) of the tree species within 36 secondary units (5 m × 5 m) were recorded. Ten of the 25 m² units were selected for randomly selecting 1 subplot (2 m × 2 m) and collecting shrub data. A microplot (1 m × 1 m) was selected in the 4 m² subplot to obtain herb data. Measures of height, percentage cover, and abundance were recorded simultaneously. The groups of trees and treelets were defined as DBH of ≥5 cm and high of <1.3 m, respectively. Plant nomenclature was normalized according to the Chinese Academy of Sciences [74].

### 2.3. Explanatory Variables

We selected 7 variables to estimate their contribution to plant diversity, some of which were those selected in previous studies [63,75,76]. The annual mean temperature (AMT, °C), annual precipitation (AP, mm), and monthly (July–August) mean sunshine duration (SSD, hours) data were derived from the China Meteorological Administration website [77] for our plots. The topographical variables (elevation, m; slope, °) were obtained during the data investigation. We used the standard deviation of DBH (DBH (SD)), as a biotic predictor, to represent the difference in ontogenetic tree stage. The forest types (FT)s were clustered by the important value (IV) of trees. The IV of trees (DBH > 5 cm) was used to classify different forest types from the sampled units, as the forests were dominated by one or more tree species with a competitive advantage [78]. This method was helpful for highlighting the characteristics of the community based on the dominant trees. The forest types were labeled according to the mean IV of major trees in the same category. It was calculated as [60,79]:

$$IV = (\text{relative abundance} + \text{relative frequency} + \text{relative height} + \text{relative DBH})/4 \tag{1}$$

### 2.4. Data Analysis

All descriptive statistics were calculated in R programming language (University of Auckland, NZ) [80]. In the cluster analysis, the 'ward. d2' method was used with the hclust function (R package 'stats'). We used IV of trees as input data for cluster analysis. The species richness, Shannon diversity index and Simpson dominance index were counted with the R 'vegan' package. We quantified the relative contribution of these 7 drivers mentioned above to species richness with the traditional gradient boosting model (GBM) [81]. This model was found to be reliable in ecology due to its flexibility and the explanatory variable selection cross-validation approach [82]. Moreover, it can fit the nonlinear relationship continuously between diversity and factors. The species richness and the factors of the 3160 plots were used in the model. The model was fitted using the gbm function step in the 'dismo' package [83].

The GBM is considered advantageous in being able to accommodate missing values and different types of predictors, immunize the impact of extreme outliers, and fit the interactions between predictors [84]. The GBM contains multiple individual models composed of simple classification regression trees and can partition the observations of response variables with similar values into groups based on a series of binary rules [85]. An iterative method was used to develop the model, where trees and re-weighted data are gradually added to remedy the cases that were poorly fit by previous trees [49].

For these reasons, the GBM can be a potent tool when analyzing complicated ecological data. However, some challenges were encountered during its use, such as (I) the contribution due to the interaction of effects whose potential complexity increases along with the size of individual trees and which is also difficult to detect; (II) the GBM might be over-fitted to the trained data due to trees being continuously added, until all the observations were perfectly explained; and (III) the optimal number of trees should be identified to maximize the accuracy of the model. When fitting this model, 2 main parameters should be stipulated. Firstly, the learn rate regulates the weight of individual trees, where smaller values correspond to more trees being added [80]. Secondly, tree complexity controls the interactional depth of each tree, where a depth of 1 indicates that a single node exists in each tree and a depth of 2 indicates 2 nodes that allow bidirectional interaction, and so on [48]. The other parameters remain unchanged.

We fitted three GBMs with a learn rate value of 0.001 and a tree complexity of 1, 2, and 5, respectively. The deviation interpretation and prediction performance of the GBMs was improved by adding simple interactions. Compared with the non-interaction models, the importance of interactions between the environmental variables was demonstrated [49]. We used 'gbm', a script that partitions the importance of variables in all the trees to the model fit, to estimate the contribution of each

predictor [80,83] (for instance: AMT, AP, SSD, elevation, slope, DBH (SD) and (FT). Finally, we summed the average value of relative contribution in the three models for the functional groups.

## 3. Results

### 3.1. Forest Type Cluster

Cluster analysis identified six blocks across the sample units (Figure 2). The six forest types were *Larix gmelinii* forest (LGF), *Larix olgensis* forest (LOF), mixed broadleaved forest (MBF), *Populus davidiana* and *Betula platyphylla* forest (PBF), mixed *Pinus koraiensis*-broadleaved forest (PKF), and mixed *Pinus tabulaeformis*-broadleaved forest (PTF) (Table 1).

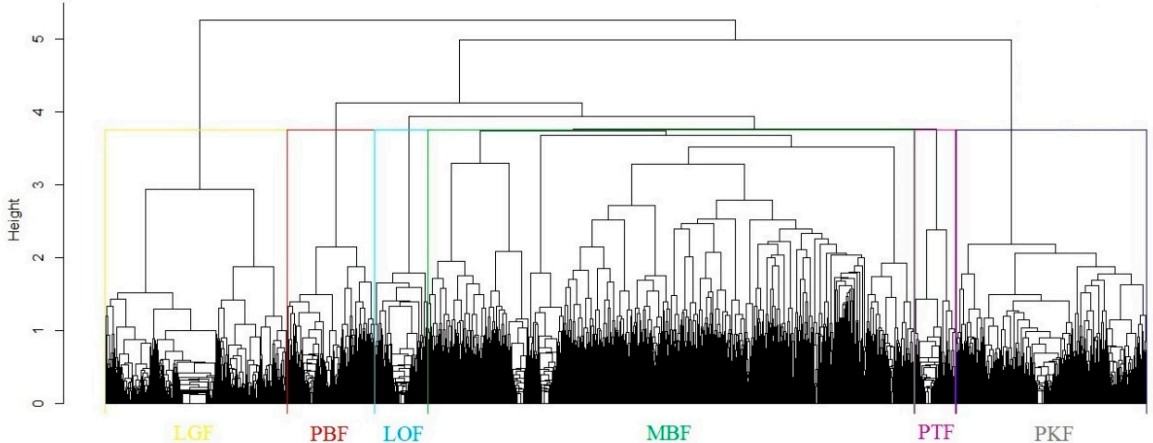

**Figure 2.** Ward's minimum variance clustering. Among the six forest types, *Larix gmelinii* forest (LGF) and *Larix olgensis* forest (LOF) are coniferous forest; mixed broadleaved forest (MBF) and *Populus davidiana* and *Betula platyphylla* forest (PBF) are broadleaved forest and mixed broadleaved forest (MBF) developed earlier; *Pinus koraiensis*-broadleaved forest (PKF) and *Pinus tabulaeformis*-broadleaved forest (PTF) are theropencedrymion forest.

**Table 1.** Mean IV of top five trees.

| Forest Type | Abbr. | FT | IV | | | | |
|---|---|---|---|---|---|---|---|
| *Larix gmelinii* forest | LGF | A | LG 0.58 | BP 0.17 | PS 0.03 | PD 0.02 | PK2 0.02 |
| *Larix olgensis* forest | LOF | B | LO 0.65 | UD 0.04 | QM 0.04 | PK1 0.04 | BP 0.03 |
| Mixed broadleaved forest | MBF | C | QM 0.26 | TA 0.07 | AM 0.06 | JM 0.05 | BD 0.04 |
| *Populus davidiana* and *Betula platyphylla* forest | PBF | D | BP 0.39 | PD 0.21 | LG 0.04 | QM 0.04 | UD 0.04 |
| *Pinus koraiensis* broadleaved forest | PKF | E | PK1 0.29 | AN 0.09 | TA 0.06 | AM 0.05 | BP 0.05 |
| *Pinus tabulaeformis* broadleaved forest | PTF | F | PT 0.68 | QM 0.10 | FR 0.03 | AS 0.03 | UM 0.02 |

Note: FT is the codes of forest type and will be used in gradient boosting model (GBM). The dominant tree species in MBF are all hardwood species and this group was called mixed broadleaved forest, although the IV of QM was the highest. The main tree species were *Betula platyphylla* (BP), *Betula dahurica* (BD), *Populus davidiana* (PD), *Larix olgensis* (LO), *Larix gmelinii* (LG), *Pinus sylvestris* (PS), *Pinus koraiensis* (PK1), *Pinus tabuliformis* (PT), *Picea koraiensis* (PK2), *Abies nephrolepis* (AN), *Quercus mongolica* (QM), *Ulmus davidiana* (UD), *Tilia amurensis* (TA), *Acer mono* (AM), *Juglans mandshurica* (JM), *Fraxinus rhynchophylla* (FR), *Armeniaca sibirica* (AS), and *Ulmus macrocarpa* (UM).

### 3.2. Spatial Distribution of Forest Types

The spatial distribution of the six forest types, as the results of clustering with IV, is depicted in Figure 3. Different forest types had their own specific distribution areas, especially LGF, LOF, and PTF.

The distribution range of MBF and PBF was the widest and that of PTF was the smallest. LOF, MBF, PBF, and PKF intersected to a certain extent, indicating that they had a nested structure.

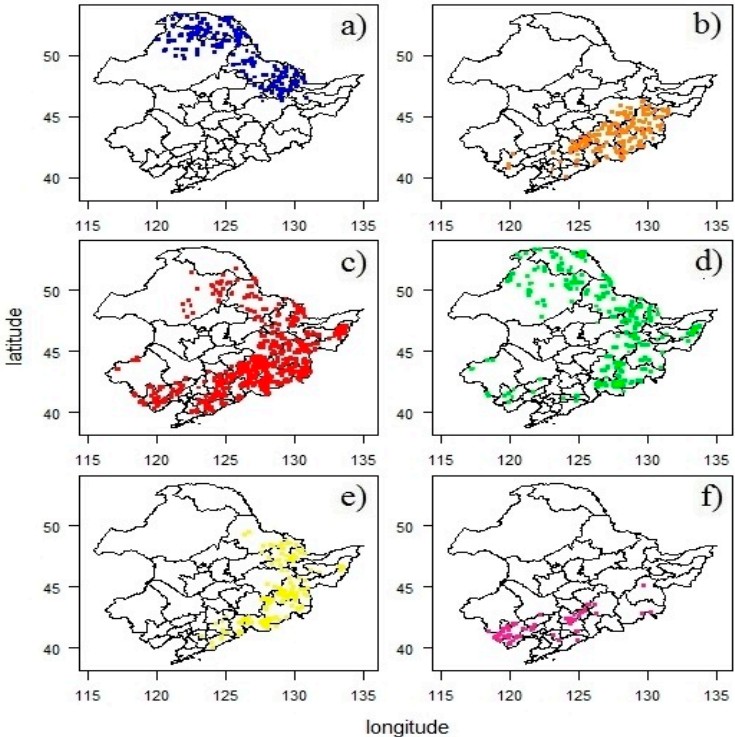

**Figure 3.** Spatial distribution of the six forest types in the region. (**a**) *Larix gmelinii* forest, (**b**) *Larix olgensis* forest, (**c**) Mixed broadleaved forest, (**d**) *Populus davidiana* and *Betula platyphylla* forest, (**e**) *Pinus koraiensis* broadleaved forest, and (**f**) *Pinus tabulaeformis* broadleaved forest.

### 3.3. Plant Richness in Different Groups

The total number of plant species (species richness) investigated from the 3160 sample units was 2170, and includes 224 trees, 314 shrubs, and 1632 herbs. The mean species richness of the total units was 34, with values ranging from 1 to 107. However, herbs accounted for over 50% of the community richness.

We also compared the differences of plant richness for each component based on IV clustering results. Analysis of variance showed that the species richness of different groups in each forest type were significantly different ($p \leq 0.01$). The lowest average richness of forest types in the four functional groups (community, tree, shrub and herb) was found in the LGF, and the highest values in each functional group were found in PKF and LOF.

We used the least significant difference (LSD) as the post-hoc test of the multiple comparative analysis (Figure 4). It showed that (1) in the community groups, the richness of the two coniferous forests (LGF and LOF) were significantly different from the others, whereas that between the natural forests (MBF, PKF, and PTF) and those of PBF and PTF were not; (2) in the tree groups, the LGF richness was significantly different from the others, whereas those of LOF, MBF, PKF, and the PBF, PTF were not; (3) in the shrub group, the richness of LGF and PBF were significantly different from the others, whereas that between LOF, MBF, and PTF and LOF, PKF, and PTF were not; and (4) in the herb group, the richness was similar to that of the community.

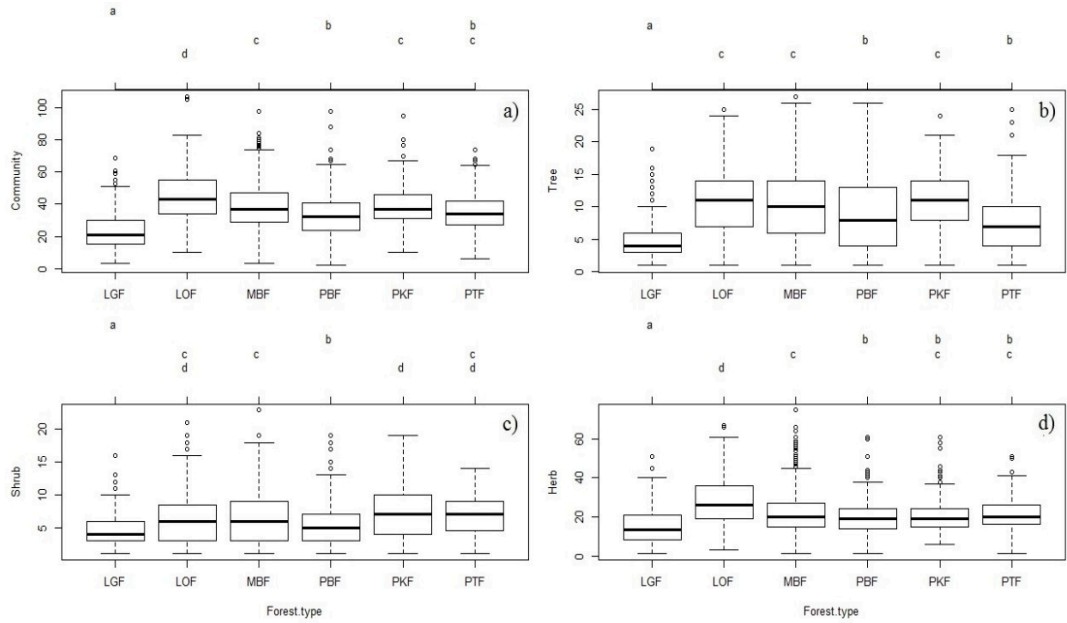

**Figure 4.** Mean richness of function groups in different forest types. (**a**) total richness, (**b**) tree richness, (**c**) shrub richness, and (**d**) herb richness. The same letters indicate not significant differences between forest types.

### 3.4. Importance of Variables

The different factors accounted for 37% to 55% of the plant richness and varied according to functional group. Among the explanatory variables, AMT and SSD contributed the most to the variation in tree, total, shrub, herb, and treelet richness, ranging from 36.2% to 61.1% and 11% to 28.5%, respectively. DBH (SD) and FT accounted for 32.5% and 33.7%, and 19.6% and 19.3% of the variation in the Shannon diversity index and Simpson dominance index, respectively. Elevation and slope contributed less to plant diversity (Table 2).

**Table 2.** Average contribution of variables to plant richness fitted in the three gradient boosting models (GBMs) with varying tree complexity (1, 2, and 5).

| Diversity | Contribution of Variables (%) | | | | | | | Parameters | |
|---|---|---|---|---|---|---|---|---|---|
| | AMT | SSD | AP | DBH (SD) | FT | EL | SL | TC | $R^2$ |
| Tree richness | 52.39 | 11.29 | 9.61 | 17.75 | 5.50 | 2.56 | 0.89 | 2 | 0.55 |
| Shannon wiener index | 30.37 | 6.84 | 4.80 | 32.45 | 19.64 | 4.72 | 1.17 | 2 | 0.51 |
| Simpson dominance index | 24.34 | 8.20 | 8.04 | 33.73 | 19.33 | 3.38 | 3.03 | 2 | 0.43 |
| Total richness | 53.79 | 19.03 | 10.91 | 8.16 | 5.37 | 1.47 | 1.27 | 2 | 0.49 |
| Shrub richness | 36.22 | 28.48 | 8.16 | 9.48 | 7.82 | 3.89 | 5.95 | 2 | 0.37 |
| Herb richness | 47.79 | 20.51 | 13.64 | 4.26 | 6.73 | 4.85 | 2.21 | 2 | 0.39 |
| Treelet richness | 61.05 | 10.97 | 9.12 | 6.76 | 5.18 | 5.14 | 1.78 | 2 | 0.45 |

Note: AMT, annual mean temperature, °C; SSD, sunshine duration, hours; AP, annual precipitation, mm; DBH (SD), standard deviation of diameter at breast height; FT, forest type; EL, elevation (m); SL slope (°); TC, tree complexity; and $R^2$, explained proportion of total deviance.

For tree richness, AMT was most impacted, followed by DBH (SD), SSD, and AP with 52.4%, 17.8%, 11.3%, and 9.6% relative contribution, respectively (Table 2). The tree richness increased and then decreased slightly with AMT and peaked at about 5 °C (Figure 5a). The richness changed with SSD into hump-shaped, ranging from 180 to 220 h (Figure 5a). Richness increased with AP and then smoothed from 600 to 1100 mm (Figure 5a). Richness increased with DBH (SD) was similar to AP, but with higher variation (Figure 5a). The changes in the tree Shannon diversity index and Simpson

dominance index were analogous to tree richness in AMT, AP, and DBH (SD), except for the SDD, which was found to be a declining ladder type and FT peaked at class C, D, and E (Figure 5b,c).

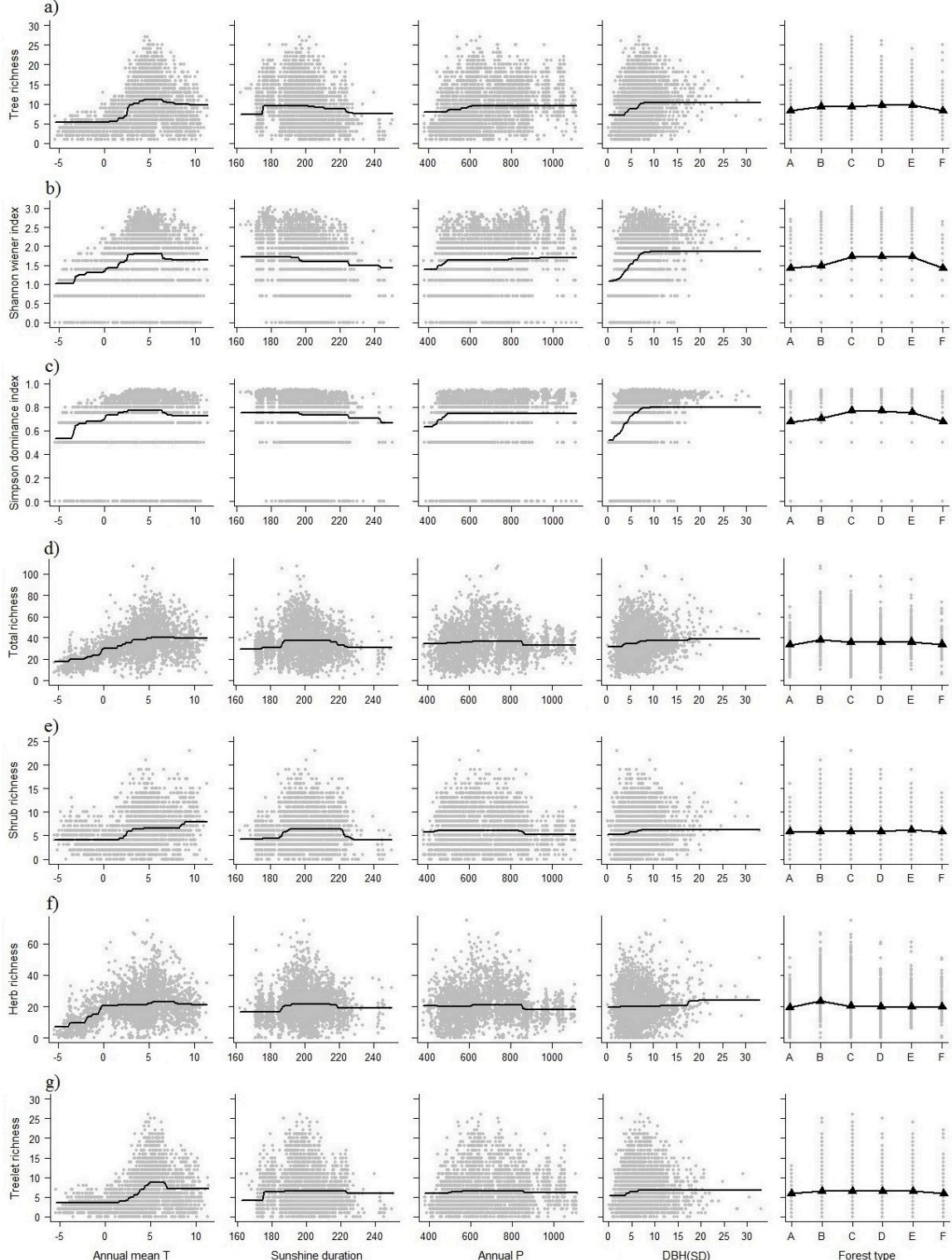

**Figure 5.** Relationships between plant richness and predictors. (**a**) Tree richness, (**b**) tree Shannon diversity index, (**c**) Simpson dominance index, (**d**) total richness, (**e**) shrub richness, (**f**) herb richness, and (**g**) treelet richness. T, temperature; P, precipitation.

AMT had the strongest influence on total richness, followed by SSD, AP, and DBH (SD) with 53.8%, 19%, 10.9%, and 8.2% relative contribution, respectively (Table 2). The total richness increased with AMT and reached a maximum from 5 to 10 °C (Figure 5d). The richness changed with SSD into hump-shaped, ranging from 190 to 210 h (Figure 5d). The optimum interval of AP was 600 to 800 mm (Figure 5d). Richness increased with DBH (SD) from 0 to 20 (Figure 5d).

For shrub richness, AMT and SSD contributed most, followed by DBH (SD) and AP with 36.2%, 28.5%, 9.5%, and 8.2% relative contribution, respectively (Table 2). The richness increased with AMT in an ascending ladder type (Figure 5e). The richness changed with SSD as hump-shaped, ranging from 190 to 220 h (Figure 5e). Richness increased with DBH (SD) and then smoothed (Figure 5e). The optimum interval of AP was 450 to 850 mm (Figure 5e). The herb richness changed with the same variables as shrub richness (Figure 5f).

AMT most impacted treelet richness, followed by SSD and AP with 61.1%, 11%, and 9.1% relative contribution, respectively (Table 2). The total richness changed with AMT similarly to tree richness and peaking at about 5 °C (Figure 5g). The richness changed with SSD in a hump shape, ranging from 180 to 220 h (Figure 5g). The changes in AP and DBH (SD) were analogous to those of shrub richness (Figure 5g).

## 4. Discussion

We aimed to verify and compare whether the contributions of explanatory variables to plant species richness differed in different functional groups. Our results show that the annual mean temperature, sunlight duration, and the tree structure strongly influenced plant richness, whereas no strong impact of the topographic factors on plant richness was observed. The forest type only strongly influenced the Shannon diversity index and the Simpson dominance index.

Numerous models between climate and plant richness have been established, indicating that temperature and precipitation significantly affect richness patterns [23,30,31,55,71,86]. Strong positive relationships between the plant richness and the annual mean temperature were observed in our results. Other studies showed that high species richness is generally found in places with high temperature along the latitude gradient [26,31]. The vascular plant diversity exhibited a significant positive correlation with precipitation in boreal forests, suggesting that diversity of wetter sites is higher than that of drier sites [29], but this was only observed in tree richness and the other groups of richness displayed a slight hump type. The intermediate disturbance hypothesis (IDH) and the niche-assembly hypothesis deemed that an intermediate environmental gradient could maintain higher diversity [47–49,87]. In boreal forests, this was consistent in the tree, shrub, and herb functional groups when experiencing disturbances [50]. The plant richness was expected to be a hump pattern along the intermediate levels of environmental stress [47].

The richness of trees and treelets increased with temperature and then decreased slightly after reaching the peak value, but this phenomenon was not obvious in other groups. As the hydrothermal condition indicates energy, we analyzed that there may be a range of saturation in the use of energy by tree layer along the latitudinal gradient before the trees with higher energy requirements formed or entered, which may indicate a redundancy of energy [11,88], as occurs with light saturation in canopy photosynthesis [89]. When the canopy reached energy saturation, it tended to benefit the dominance of thermophilic–photopic species and maintain more individuals so that excess energy could be absorbed to reduce the risk of extinction [70,90–92]. Higher species evenness could weaken the light availability of the understory, and thus reduce shade-intolerant tree species [91,93]. These factors might influence the tree community development toward a simple species composition, as the mean tree richness of PTF was lower (Figures 3f and 4b). When the tree richness reached the peak value (about 5 °C), the increased tree abundance and reduced DBH (SD), supporting this inference (Figure 6).

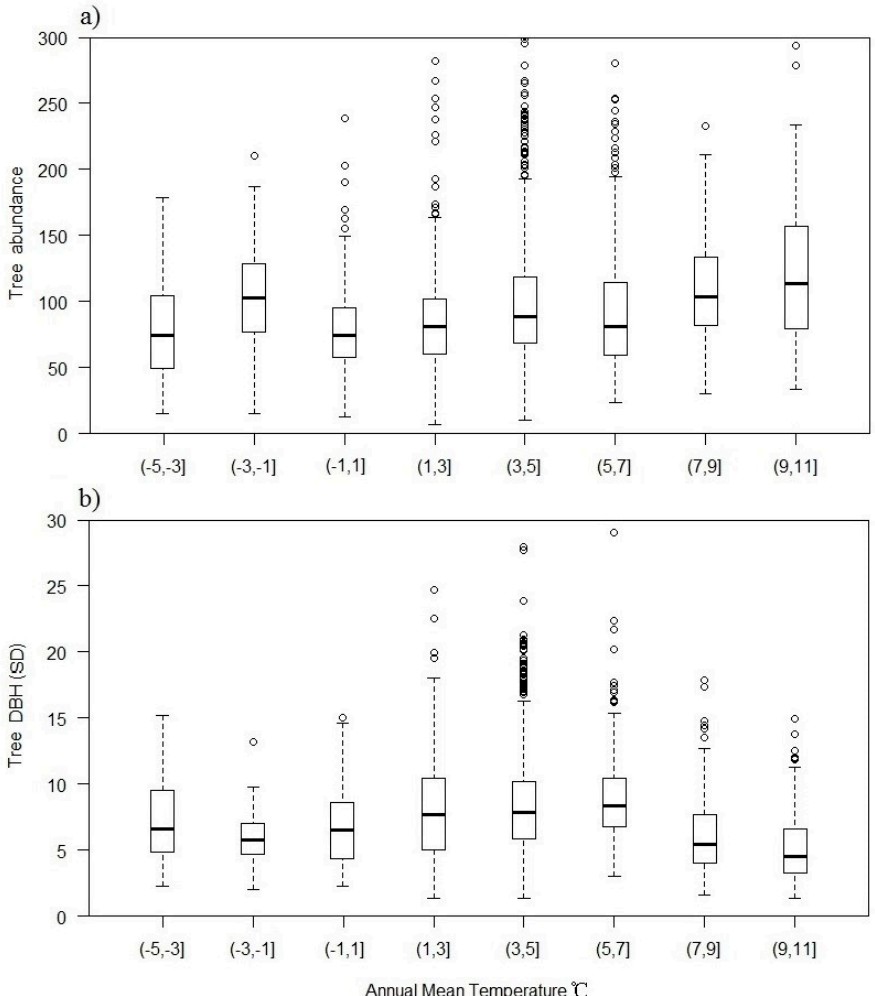

**Figure 6.** Relationships between the tree structure and annual mean temperature: (**a**) tree abundance and (**b**) tree DBH (SD).

The plant richness increased with DBH (SD). The tree diameter distribution, as one of the most commonly used indexes, has been widely used to predict forest structure [94]. Trees of different ages and sizes represented different ontogenesis and could form complex tree structure [95], thereby increasing the diversity [91]. Other studies found that the temporal changes in stand structure and plant composition in a forest ecosystem varied with the succession stage [96,97], and this change regulates the heterogeneity of light conditions through canopy closure [98,99] and edaphic features (e.g., the soil moisture content, pH, temperature, and nutrient availability) under the forest canopy [22,31,57,100–102]. An uneven forest overstory weakens the adverse effects of seedling survival [103]. Individuals at different stages of development have different energy requirements, so a layered vertical structure can maximize the energy use at different heights, thus increasing species richness.

Strong direct and indirect effects are found among plant species [104–106], and the effects between organisms are self-adaptation and belong to the biotic factors [107]. These interactions demonstrate that competition could produce positive or negative effects on individuals within community [108,109]. Therefore, we could not ignore the biotic factors in the ecological process, although the relative contribution of climate is higher than that of biological factors (stand structure and forest type) on large scale.

Our results show that the plant species richness of forest ecosystems is mainly regulated by climatic conditions on regional scale. Forest type could impact plant richness (Figure 4), but not very much (Table 2). This may attribute to that forest types, as the biotic factors, represent the difference of

community patterns [70,72], which can provide different hydrothermal conditions [23,51]. However, these contributions are small, relative to the climate on regional scale. Furthermore, plant richness of different forest types may be also affected by the regional species pool [31], especially in the mosaic pattern (Figure 3b–e).

The relative contribution of climatic variables is larger than that of topographical factors [31], which is consistent with our results. However, it does not mean there is no influence of topographic factors on plant richness, but very small relative to climate factors. As the topographic factors significantly impact the local species richness [38,39] and the latitude effect between richness-temperature [23,24], we analyze that the dominant factors of diversity at local and regional scales are topography and climate, respectively.

The contributions of environmental factors to the richness in different functional groups are heterogeneous, as is the response of the richness of each component to environmental factors. Previous researchers also found that environmental factors had various influences on plant richness in different functional groups [23]. This difference could be determined by the interaction of multiple mechanisms, but not by single mechanism. Zhang et al. [23] found that the inconsistency could be attributed to different functional groups having different natural resource requirements, which might be influenced by different development stages, stand structure, vegetation composition, light conditions, and the edaphic feature of the forest [31,61,63,75,110]. This means that different groups have their own optimum environmental range, known as the optimum ecological niche [49]. The differences in plant life forms and their life history strategies could result in this inconsistency, and contribute to the relative plant richness and the response to environmental factors [111,112].

## 5. Conclusions

Our findings illustrate that plant species richness is mainly regulated by temperature and sunlight, followed by DBH (SD) on a regional scale. Temperature and precipitation could promote plant richness, but with an upper limit. Plant richness varies with sunshine duration in an optimal range. Stand structure can regulate the relationship of tree richness and temperature along with the latitude. Different community structures would affect the utilization efficiency of vegetation to environment. Forest type could influence plant richness, but not too much relative to climatic factors. Topographic factors impact plant richness a little on large scale. The drivers influencing plant richness differed according to functional group, due to the different environmental resource requirements and the different life history strategies of plant layers. However, we found that plant diversity in a forest ecosystem is not determined by a single mechanism, but rather by a combination of environmental drivers. Our results emphasize the differences in the multiple variables used to determine species richness among groups in the temperate forest in Northeast China. In forest management, we should construct community of thermophilic–photopic trees in the canopy and shade-intolerant species in the understory. The community should better form a vertically stratified structure with different ages and sizes of individuals, which helps plants use the temperature and light, thus increasing plant diversity. To evaluate an ecological process, we should consider the interaction between different drivers, not the importance of a single mechanism only.

**Author Contributions:** conceptualization, Y.G., S.H., and Z.C.; methodology, Y.G., J.Z., and Y.F.; software, Y.G., J.Z., and S.G.; Project administration, S.H. and J.Z.; Resources, S.H.; Supervision, S.H.; Validation, Y.G., S.H., and J.Z.; Visualization, Y.G.; Writing—Original draft, Y.G.; Writing —Review & editing, Y.G., W.W., Z.C., and Y.J.; Funding acquisition, S.H., J.Z., and S.G. All authors have read and agreed to the published version of the manuscript.

**Funding:** This work was supported by the National Natural Science Foundation of China (Nos. 41575153, 41430639 and 31800413) and the National Key Research and Development Program of China (No. 2016YFA0600804).

**Conflicts of Interest:** The authors declare no conflict of interest.

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
