# Peer review of "Temperature-Dominated Driving Mechanisms of the Plant Diversity in Temperate Forests, Northeast China"

_forests, doi:10.3390/f11020227_

Round 1
Reviewer 1 Report
27.01.2020
Manuscript ID forests-710239, Forests
Temperature-dominated driving mechanisms of the plant diversity in temperate forests, Northeast China”
The authors responded and addressed the majority of my comments from first round of review. However, some points still need clarification. Below I am attaching some suggestions, of which implementation can improve flow of the manuscript.
Abstract
The main issue connected with this part that remains unclear, is to which aspects of diversity the authors refer (please see lines 21-22, for instance). Do they mean total species richness of vascular plants, including both number of species occurring in tree and herb layer, of separately? Please clarify.
L 17 We assessed the contribution…
L 31 Inter or intra-species heterogeneity of trees? Please specify.
Introduction
L 46-51 In my opinion there is too big generalization of patterns revealed by previous authors. It is not clear to which type of vegetation the authors refer here. Different climatic parameters would affect composition and structure of different plant communities in different way, but in the current version no information on these relationships is provided.
L 80-85 Set of hypotheses should be reformulated, since now it creates many biases:
For instance, while annual precipitation promotes diversity with an upper limit, the same would be linked with temperature? What means that plant diversity varies with sunshine duration in hump-shaped? Plant diversity changes with forest type – probably the hypothesis of low novelty: the fact that different forest types differ in plant species diversity is well known for a long time, e.g. deciduous vs. coniferous forest L 83 ontogenic stage of what? Please specify. Again, does the authors mean diversity of plant species existing in forest herb layer, tree layer, or both?Materials and Methods
L 94-96 The main shrubs, for instance, are: R. aureum, L. japonica, L. bicolor and E. alatus. The main herbs, in turn, are A. heterophyllum, S. japonica, B. paridiformis and O. corniculata.
L 97 Phrase “The regional climate is mountain monsoon forest of temperate…” sounds strange to me
L 121-122 SD of DBH was used as a predictor of plant diversity or response variable?
L 133 Which drivers the authors mean here?
L 158-160 This sentence sounds unclear to me. Please rewrote
L 162 …of seven predictors used in our study (here, in brackets you can list these variables)
L 163 Unclear, what mean functional groups of forest types/
L 162-163 Does it mean that for each type of forest type and each functional group of plants was performed one model with different three complexity? Anyway, on Fig 5 is presented something different, i.e. one separate model was performed for each functional group of plants, where forest type was included as a predictor.
Results
L 183 But results of hclust are presented on Fig. 2!
L 201 A description that ANOVA and multiple comparative analysis were employed in order to determine differences in species richness of different functional groups of plants between forest type should be placed in M&M section. In addition, which kind of post-hoc test implemented in multiple comparative analysis was used?
200-211 Do these two paragraphs refer to Fig. 4? 219 Table 2. In M&M (L 157-158) the authors write that the performed models with different tree complexity, but in table is presented only one threshold.L 235 What means “h” letter after the number “210”?
Discussion
I have an impression that this part of the study is still too generalized:
The authors did not find significant relationships of vegetation parameters with topographic factors – this is extremely interesting why there was no influence, while many other studies reveal a strong gradient of vegetation that changes along increasing elevation or changing topography. In current version of manuscript, this aspect is partly discussed from line 305, but should be emphasized more. I recommend some more detailed discussion of the patterns revealed by the authors during their work with results of different researches regarding different forest types (e.g. not only boreal, as referring to this forest type is predominating now).L 285 It is still unclear, what presents Fig. 6, for detecting what it was performed, and why this is a part of Discussion, not results. In addition, what mean numbers in round/square brackets on x-axis?
L 297 are not only a response
Author Response
Response to Reviewer 1 Comments
“The authors responded and addressed the majority of my comments from first round of review. However, some points still need clarification. Below I am attaching some suggestions, of which implementation can improve flow of the manuscript.”
Abstract
Point 1: “The main issue connected with this part that remains unclear, is to which aspects of diversity the authors refer (please see lines 21-22, for instance). Do they mean total species richness of vascular plants, including both number of species occurring in tree and herb layer, of separately? Please clarify.
Response 1: Dear reviewer, thank you for your detailed opinions. The tree, shrub and herb are separated completely according to the Chinese Academy of Sciences (as reference 74) to make sure they are not crossed. We have modified it to: (L19-22) We investigated about 0.986 million plants from 3160 sites to quantify the impact of annual mean temperature, sunshine duration, annual precipitation, standard deviation of diameter at breast height, and forest type on richness of vascular plants (total species, tree, treelet, shrub and herb, separately) using the gradient boosting model.
Point 2: “L 17 We assessed the contribution…”
Response 2: Dear reviewer, thank you for your detailed opinions. We have modified it to: (L17-18) In this study, we assessed the contribution of these drivers and how they affect the vascular plant richness…..
Point 3: “L 31 Inter or intra-species heterogeneity of trees? Please specify.”
Response 3: Dear reviewer, thank you for your detailed opinions. We have modified it to: (L31-32) The relationship between tree richness and annual mean temperature with latitudinal effect could be affected by the differences in number and size of tree individuals…..
Introduction
Point 4: “L 46-51 In my opinion there is too big generalization of patterns revealed by previous authors. It is not clear to which type of vegetation the authors refer here. Different climatic parameters would affect composition and structure of different plant communities in different way, but in the current version no information on these relationships is provided.”
Response 4: Dear reviewer, thank you for your detailed opinions. We have modified it to: (L50-52) Climatic gradient is considered as the main abiotic factor controlling large-scale species diversity [21–24]. Strong positive relationships between climate and plant richness are mainly impacted by temperature variations as well as latitude effects [23, 25,26].
Point 5: “L 80-85 Set of hypotheses should be reformulated, since now it creates many biases:
For instance, while annual precipitation promotes diversity with an upper limit, the same would be linked with temperature? What means that plant diversity varies with sunshine duration in hump-shaped? Plant diversity changes with forest type – probably the hypothesis of low novelty: the fact that different forest types differ in plant species diversity is well known for a long time, e.g. deciduous vs. coniferous forest. L 83 ontogenic stage of what? Please specify. Again, does the authors mean diversity of plant species existing in forest herb layer, tree layer, or both?”
Response 5: Dear reviewer, thank you for your detailed opinions. We agree with your viewpoint. Different forest types have different community structures and species composition. While different forest types may have the same or similar richness at nearby sites. Here, we want to express that this results need model simulate by a mass of data on large scale (as fig. 5). We separate the plants into tree, shrub, herb and treelet. The seedings of tree are divided into treelet, not into herb. We have modified it to: (L84-89) We hypothesized that (I) plant richness increases with annual mean temperature and annual precipitation, but have an upper limit; (II) plant richness varies with sunshine duration in hump-shaped relationships; (III) the greater the standard deviation of DBH, the higher the plant richness; (IV) forest type can impact plant richness, but not very much relative to other factors; (V) topographic factors influence plant richness a little on large scale; and (VI) the relative contribution of the drivers to plant diversity and the responses vary with functional groups on a regional scale.
Materials and Methods
Point 6: “L 94-96 The main shrubs, for instance, are: R. aureum, L. japonica, L. bicolor and E. alatus. The main herbs, in turn, are A. heterophyllum, S. japonica, B. paridiformis and O. corniculata.”
Response 6: Dear reviewer, thank you for your detailed opinions. We have modified it to: (L96-98) The main shrubs, for instance, are: Rhododendron aureum, Lonicera japonica, Lespedeza bicolor, Euonymus alatus et al. The main herbs, in turn, are Arisaema heterophyllum, Saussurea japonica, Brachybotrys paridiformis, Oxalis corniculate et al.
Point 7: “L 97 Phrase “The regional climate is mountain monsoon forest of temperate…” sounds strange to me”
Response 7: Dear reviewer, thank you for your detailed opinions. We have modified it to: (L98-99) The region is mountain monsoon forest of temperate continental climate zone and characterized by four distinctive seasons.
Point 8: “L 121-122 SD of DBH was used as a predictor of plant diversity or response variable?”
Response 8: Dear reviewer, thank you for your detailed opinions. SD of DBH was a biotic factor (predictor). We have modified it to: (122-123) We used the standard deviation of DBH (DBH (SD)), as a biotic predictor, to represent the difference in ontogenetic tree stage.
Point 9: “L 133 Which drivers the authors mean here?”
Response 9: Dear reviewer, thank you for your detailed opinions. We have modified it to: (134-135) We quantified the relative contribution of these 7 drivers mentioned above to species richness with the traditional gradient boosting model (GBM) [81].
Point 10: “L 158-160 This sentence sounds unclear to me. Please rewrote
L 162 …of seven predictors used in our study (here, in brackets you can list these variables)”
Response 10: Dear reviewer, thank you for your detailed opinions. We have modified it to: (159-161) The deviation interpretation and prediction performance of the GBMs was improved by adding simple interactions. Compared with the non-interaction models, the importance of interactions between the environmental variables was demonstrated [49]. (L163) …of each predictor [80,83] (for instance: AMT, AP, SSD, elevation, slope, DBH(SD) and FT).
Point 11: “L 163 Unclear, what mean functional groups of forest types/
L 162-163 Does it mean that for each type of forest type and each functional group of plants was performed one model with different three complexity? Anyway, on Fig 5 is presented something different, i.e. one separate model was performed for each functional group of plants, where forest type was included as a predictor.”
Response 11: Dear reviewer, thank you for your detailed opinions. We have modified it to: (L163-164) Finally, we summed the average value of relative contribution in the three models for the functional groups.
Results
Point 12: “L 183 But results of hclust are presented on Fig. 2!”
Response 12: Dear reviewer, thank you for your detailed opinions. The hclust of tree IV is the basics of Table 1, Fig 3 and Fig 4. Table 1 is the specific results of Fig 2 and we name the forest type according to it. Fig 3 is the the spatial distribution of forest types and it shows the difference of their distribution. Fig 4 represents the difference of plant richness of forest type. They all indicate that the hclust of tree IV is reliable.
Point 13: “L 201 A description that ANOVA and multiple comparative analysis were employed in order to determine differences in species richness of different functional groups of plants between forest type should be placed in M&M section. In addition, which kind of post-hoc test implemented in multiple comparative analysis was used?”
Response 13: Dear reviewer, thank you for your detailed opinions. Figure 4 can not only illustrate the difference of plant richness in clustering, but also intuitively compare the average richness of each forest type. So we choose this expression. The post-hoc test we used was LSD. We have modified it to: (L207-208) We used the least significant difference (LSD) as the post-hoc test of the multiple comparative analysis (Figure 4). It showed that….
Point 14: “200-211 Do these two paragraphs refer to Fig. 4? 219 Table 2. In M&M (L 157-158) the authors write that the performed models with different tree complexity, but in table is presented only one threshold.
L 235 What means “h” letter after the number “210”?”
Response 14: Dear reviewer, thank you for your detailed opinions. Yes, the two paragraphs refer to Fig. 4 and we have modified it to: (L207-208) We used the least significant difference (LSD) as the post-hoc test of the multiple comparative analysis (Figure 4). We did use different tree complexity (1, 2 and 5) in the models, but we finally summed the average of them in Table 2. We illustrate this in L164-165 and the title of Table 2 (L222). “h” is the unit of sunshine duration and we have modified it to “hour” (L247).
Discussion
Point 15.1: “I have an impression that this part of the study is still too generalized: The authors did not find significant relationships of vegetation parameters with topographic factors – this is extremely interesting why there was no influence, while many other studies reveal a strong gradient of vegetation that changes along increasing elevation or changing topography. In current version of manuscript, this aspect is partly discussed from line 305, but should be emphasized more. ”
Point 15.2: “I recommend some more detailed discussion of the patterns revealed by the authors during their work with results of different researches regarding different forest types (e.g. not only boreal, as referring to this forest type is predominating now).”
Response 15.1: Dear reviewer, thank you for your detailed opinions. In our results, the influence of topographic factors on plant richness is very small relative to climate factors on regional scale, but we do not mean no influence. We agree with We have modified it to: (L312-317) The relative contribution of climatic variables is larger than that of topographical factors [31], which is consistent with our results. However, it does not mean there is no influence of topographic factors on plant richness, but very small relative to climate factors. As the topographic factors significantly impact the local species richness, we analyze that the dominant factors of diversity at local and regional scales are topography and climate, respectively.
Response 15.2: Dear reviewer, thank you for your detailed opinions. Forest type could impact plant richness (Figure 4), but not dominant relative to climatic factors (Table 2). We have modified it to: (L305-311) Our results show that the plant species richness of forest ecosystems is mainly regulated by climatic conditions on regional scale. Forest type could impact plant richness (Figure 4), but not very much (Table 2). This may attribute to that forest types, as the biotic factors, represent the difference of community patterns [70,72], which can provide different hydrothermal conditions [23,51]. But these contribution are small relative to the climate on regional scale. Furthermore, plant richness of forest types may be also affected by the regional species pool [31], especially in the mosaic pattern (Figure 3b,c,d,e).
Point 16: “L 285 It is still unclear, what presents Fig. 6, for detecting what it was performed, and why this is a part of Discussion, not results. In addition, what mean numbers in round/square brackets on x-axis?”
Response 16: Dear reviewer, thank you for your detailed opinions. We find that the richness of trees and treelets increased with temperature and then decreased slightly after reaching the peak value and analyzed this phenomenon in terms of energy utilization and community structure. This is also one of our findings that tree structure affects the latitude effect of diversity and Fig. 6 proves this. So we put it in the discussion, not in the results. ‘(’ is open interval and means ‘>’. ‘]’ is closed interval and means ‘≤’. i.e. (9,11] means: 9℃ < temperature ≤ 11℃
Point 17: “L 297 are not only a response”
Response 17: Dear reviewer, thank you for your detailed opinions. We have modified it to: (L299) Strong direct and indirect effects are found among plant species [104-106], and the effects between organisms are self-adaptation and belong to the biotic factors [107]
Reviewer 2 Report
Dear Authors,
I have reviewed the re-submission of the paper "Temperature-dominated driving mechanisms of the plant diversity in temperate forests, Northeast China". The aims of the paper are germane with Forests topic, in this form of article fits with the international scientific standards. The paper is written with an appropriate English level. The contribution of this paper to the scientific knowledge is good. In my opinion, there are only few flaws, and I suggest to improve the paper following the suggestions showed in the comments below.
I understand your reply and they are clear, but please it is necessary to detail your statement also in the paper. Please put a specific sentence that highlight the unmanagement situation of these forests and if possible a brief history related to the management sytuation of the past (last treatment and what kind).
And again please I suggest for a better understanding of the findings, in the conclusions to add also a synthetic point by point conclusions referred to the detailed aims.
Author Response
Response to Reviewer 2 Comments
“I have reviewed the re-submission of the paper "Temperature-dominated driving mechanisms of the plant diversity in temperate forests, Northeast China". The aims of the paper are germane with Forests topic, in this form of article fits with the international scientific standards. The paper is written with an appropriate English level. The contribution of this paper to the scientific knowledge is good. In my opinion, there are only few flaws, and I suggest to improve the paper following the suggestions showed in the comments below.”
Point 1: “I understand your reply and they are clear, but please it is necessary to detail your statement also in the paper. Please put a specific sentence that highlight the unmanagement situation of these forests and if possible a brief history related to the management sytuation of the past (last treatment and what kind).”
Response 1: Dear reviewer, thank you for your detailed opinions. We have modified it to: (L75-77) The mountain forest in Northeast China, as one of the most species-rich temperate ecosystems, has been influenced by natural factors over the last few decades. All the sample plots we investigated are unmanaged and far away from the road.
Point 2: “And again please I suggest for a better understanding of the findings, in the conclusions to add also a synthetic point by point conclusions referred to the detailed aims.”
Response 2: Dear reviewer, thank you for your detailed opinions. We have modified it to: (L330-340) Our findings illustrate that plant species richness is mainly regulated by temperature and sunlight, followed by DBH (SD) on a regional scale. Temperature and precipitation could promote plant richness, but with an upper limit. Plant richness varies with sunshine duration in an optimal range. Stand structure can regulate the relationship of tree richness and temperature along with the latitude. Different community structure would affect the utilization efficiency of vegetation to environment. Forest type could influence plant richness, but not too much relative to climatic factors. Topographic factors impact plant richness a little on large scale. The drivers influencing plant richness differed according to functional group, due to the different environmental resource requirements and the different life history strategies of plant layers. However, we found that plant diversity in a forest ecosystem is not determined by a single mechanism, but rather by a combination of environmental drivers. …...
Round 2
Reviewer 1 Report
Authors repsonded to all my comments. I have only four minor comments:
L. 81. What about other communication routes in the surrounding of the forests studied? In addition, it would be useful to mention that other human impacts are also low because of huge distance from the study area, e.g. urbanization, agriculture and industry (aren't they?). In the same line you say that "all the sample plots we investigated are unmanaged...", but in Results (L. 234-235) you write that LGF and LOF were planted. In this light, please clarify, whether the forests surveyed are under recent (or historical) impact of forest management. L. 246 There is no specification in M&M that Shannon and Simpson indexes were calculated. I guess you used vegan package (as for species richness)? L. 257-258. Unclear statement. Please rewrite L 266. Richness increased monotonically
Author Response
Response to Reviewer 1 Comments
“Authors repsonded to all my comments. I have only four minor comments:”
Point 1: “L.81. What about other communication routes in the surrounding of the forests studied? In addition, it would be useful to mention that other human impacts are also low because of huge distance from the study area, e.g. urbanization, agriculture and industry (aren't they?).”
Response 1: Dear reviewer, thank you for your detailed opinions. Yes, they are. We have modified it to: (L78-79) All the sample plots we investigated are unmanaged and far away from the road, urbanization, agriculture and industry.
Point 2: “In the same line you say that "all the sample plots we investigated are unmanaged...", but in Results (L. 234-235) you write that LGF and LOF were planted. In this light, please clarify, whether the forests surveyed are under recent (or historical) impact of forest management.”
Response 2: Dear reviewer, thank you for your detailed opinions. We have modified it to: (L208) …the richness of the two coniferous forests (LGF and LOF) … (L76-78) The mountain forest in Northeast China, as one of the most species-rich temperate ecosystems, are influenced by natural factors over the last few decades but part of them had been impacted by history eighty years ago.
Point 3: “L. 246 There is no specification in M&M that Shannon and Simpson indexes were calculated. I guess you used vegan package (as for species richness)? L. 257-258.”
Response 3: Dear reviewer, thank you for your detailed opinions. Yes, they were. We have modified it to: (L135-136) The species richness, Shannon diversity index and Simpson dominance index were counted with the R ‘vegan’ package.
Point 4: “Unclear statement. Please rewrite L 266. Richness increased monotonically.”
Response 4: Dear reviewer, thank you for your detailed opinions. We have modified it to: (L241) Richness increased with DBH (SD) from 0 to 20 (Figure 5d).
This manuscript is a resubmission of an earlier submission. The following is a list of the peer review reports and author responses from that submission.
Round 1
Reviewer 1 Report
Dear authors,
I believe that the manuscript titled "Temperature-dominated driving mechanisms of the plant diversity in temperate forests, Northeast China" is very well drafted, methods are exhaustively explained, and correctly applied to achieve the objectives of the study. The topic is also of the interest of this journal. I appreciate the efforts of the authors to obtain such amount of data, that have been well statistically analyzed, and well discused, from my point of view. The manuscript is also very well structured, and the written exppresion is correct, I think that it is easy to read and understand by the readers and users of this journal.
For that reasons, I recommend to accept this paper to be published in present form.
Author Response
Dear reviewer,
Thank you very much for your approval. We will try our best to make better modifications.
Reviewer 2 Report
19.12.2019
Manuscript ID forests-662938, Forests
Temperature-dominated driving mechanisms of the plant diversity in temperate forests, Northeast China”
Gu et al. investigated the role of abiotic factors (e.g. temperature, precipitation, insolation) in shaping patterns of plant species diversity in forest vegetation of temperate climate zone, considering the large spatial scale. While I strongly appreciate a big effort on data collection (more than 3100 study sites) and overall concept of the study, some parts of the manuscript require substantial reworking, e.g. by providing more detailed overview of previous literature focussing on the same or similar topic in introduction, better description of statistical methods and more deep discussion of the role of environmental factors shaping plant species assembling in forest communities. The application of these amendments, in my opinion, would significantly emphasize the novelty of the study, as well as better addressing the patterns revealed in the context of ecological mechanisms influencing diversity of forest vegetation.
Abstract
My suggestion is to avoid using abbreviations in abstract.
L18 …the richness of functional groups of plants in temperate forest…
Introduction
My main concern is that the authors should emphasize more the novelty of their study. In some places they claim that there is no studies investigating impacts of abiotic factors on vegetation in large spatial scale that sometimes is not true.
For instance, in L. 42-43 the authors suggest that there is insufficient amount of data on factors shaping plant species composition in large spatial scale, however this is well known for both spatial and temporal scale for European temperate forests. See Bernhardt-Romermann et al. 2015. Global Change Biology. doi: 10.1111/gcb.12993. Further, in L. 58-59 the authors state that little is known how topography affect richness and diversity of plants that may be another conjecture (at least focusing on grassland vegetation). See:
Casas C, Ninot JM (2003) Correlation between species composition and soil properties in the pastures of Plana de Vic (Catalonia, Spain). Acta Bot. Barc. 49:291-310.
BaÅŸnou C, Pino J, Šmilauer P (2009) Effect of grazing on grasslands in the Western Romanian Carpathians depends on the bedrock type. Preslia 81(2):91-104
Sanaei A, Li M, Ali A (2019) Topography, grazing, and soil textures control over rangelands’ vegetation quantity and quality. Science of the Total Environment 697, 1341523, doi: 10.1016/j.scitotenv.2019.134153
In addtion, there is no information on vegetation type, for which the effects of environmental factors on plant species diversity are well recognized by previous authors.
L47-48 Here I would expect the opposite tendency, i.e. if temperature is higher (e.g. in tropical rain forest), then species richness and diversity is expected to be higher than in cooler regions (e.g. boreal taiga forests).
L50-52 Please provide more details on how insolation can affect primary production.
L63 Unclear what means “given the shading effect of canopy”.
L81-82 Here the authors provided one sentence on the importance of their study for forest management and conservation, but this aspect should be discussed here in concrete, i.e. how the knowledge on abiotic drivers shaping compositional patterns of forest vegetation would contribute to better understanding ecological mechanisms responsible for species assembling and why these results can be important for forestry and nature conservation.
Materials and Methods
L85-91 Please provide more details on forest vegetation in the study site. Is this the mountain monsoon forest of temperate climate zone?
L106-117 Add units in which predictors you used were measured.
L110-111 Not clear why SD of DBH rather than simply DBH was used to characterize tree stand structure.
L119-150 There is unclear which analysis was used for revealing of which pattern. Focusing on clustering and Fig. 2, one may suppose that this analysis was employed for determination of different forest types, but something different is mentioned in Methods (L. 112-113; important value of tress). Did the authors used IVs as input data for clustering performance? Please specify. Similar issue is linked with GBM. In two paragraphs (L. 126-143) the authors provided detailed description of GBM and its advantages in ecological modeling. However, I missed information on the linkage between employment of GBM and data used in the study. Moreover, focusing on Fig. 4 and description in L. 185-188, it suggests that the authors used an ANOVA to describe differences between mean richness and diversity of different functional groups of plants across different types of forest communities, but there is no information on this in methods. Correspondingly, the same concern is connected with a post-hoc analysis used to use between which forest type there were significant differences.
Results
L161. Table 1 is not self-explaining. Please provide what mean abbreviations used here. I suppose that abbreviations in fourth column are forest types, but what means FT?
L175. Similar as in case of Table 1. Figure should be self-explanatory, in order to have better orientation in the text. To achieve this, maybe useful it would be to provide at least provide that for full names of forest vegetation type see Figure 2?
L183. In results there is no reference to Fig. 4 (the same with Fig. 6). Please specify that boxplots with the same letters indicate not significant differences between vegetation parameters and forest type, according to the multiple comparative analysis results.
L228. In order to make better orientation in the text, maybe you could add the percentage of average contribution of each predictor in explaining the variability of response variables?
Discussion
In the whole section there is a great emphasis on the role of abiotic factors driving richness and diversity of forest communities. However, no information on impacts of biotic factors on shaping species assembling is mentioned here, that can be equally important factors in explaining differences in species composition across different forest types. Thus, it would be grateful to see how niche processes (competition and niche partitioning) may shape species richness and diversity. I know that this was not the aim of this study, but one additional paragraph, where the authors would discuss potential interactions between both abiotic and biotic factors, and how this complex of environmental drivers may result in shift in plant species composition may be useful in order to more detailed discussion of the patterns revealed (see “integrated community concept”; Lortie, C. J., Brooker, R. W., Coler, P., Kikvidze, Z., Michalet R., Pugnaire F. I., & Callaway, R. M. 2004). Rethinking plant community theory. Oikos 107(2), 433–438). In this light, my another suggestion is to add some explanations (maybe in introduction), why the authors focused only on abiotic factors and completely excluded biotic drivers. Was it because previous studies found habitat filters as the most important drivers of forest composition? Also, ecological meaning of each predictor found to be significantly influencing forest diversity should be provided here in details.
L245-246 But this contradicts the statement provided in introduction (L. 47-49).
L273 influencing the diversity?
Conclusions
Some information on applicability of the patterns revealed in practice (e.g. forest management and nature conservation) may be put here or as the last paragraph of the discussion (see my comments from introduction L. 81-82).
Author Response
Response to Reviewer 2 Comments
Abstract
Point 1: “My suggestion is to avoid using abbreviations in abstract. L18 …the richness of functional groups of plants in temperate forest…”
Response 1: Dear reviewer, thank you for your detailed opinions. We have modified it to: (L21-30) We investigated about 0.986 million plants from 3160 sites to quantify the impact of annual mean temperature, sunshine duration, annual precipitation, standard deviation of diameter at breast height, and forest type on plant diversity using the gradient boosting model. The results show that annual mean temperature had the strongest impact on plant richness. The tree richness peaked at intermediate annual mean temperature and sunshine duration and increased with annual precipitation. The Shannon diversity index and Simpson dominance index increased with annual precipitation and standard deviation of diameter at breast height, decreased with sunshine duration, and peaked at intermediate annual mean temperature and forest type. The total richness and understory richness increased with annual mean temperature and standard deviation of diameter at breast height and peaked at intermediate sunshine duration and annual precipitation. A comprehensive mechanism was found to regulate the plant diversity in forest ecosystems. The relationship between tree richness and annual mean temperature with latitudinal effect could be affected by the heterogeneity of trees, indicating that plant diversity varies with the utilization of energy.
Introduction
Point 2: “My main concern is that the authors should emphasize more the novelty of their study. In some places they claim that there is no studies investigating impacts of abiotic factors on vegetation in large spatial scale that sometimes is not true.
For instance, in L. 42-43 the authors suggest that there is insufficient amount of data on factors shaping plant species composition in large spatial scale, however this is well known for both spatial and temporal scale for European temperate forests. See Bernhardt-Romermann et al. 2015. Global Change Biology. doi: 10.1111/gcb.12993.”
Response 2: Dear reviewer, thank you for your detailed opinions. Here, we want to express that previous studies have only explored the correlation between diversity and factors, while few studies explore how the diversity varied with factors continuously. We have modified it to: (L44-45) However, the process which the species composition and diversity forest varying with factors continuously remain ambiguous when considering large spatial scales.
Point 3: “Further, in L. 58-59 the authors state that little is known how topography affect richness and diversity of plants that may be another conjecture (at least focusing on grassland vegetation). See:
Casas C, Ninot JM (2003) Correlation between species composition and soil properties in the pastures of Plana de Vic (Catalonia, Spain). Acta Bot. Barc. 49:291-310.
BaÅŸnou C, Pino J, Šmilauer P (2009) Effect of grazing on grasslands in the Western Romanian Carpathians depends on the bedrock type. Preslia 81(2):91-104
Sanaei A, Li M, Ali A (2019) Topography, grazing, and soil textures control over rangelands’ vegetation quantity and quality. Science of the Total Environment 697, 1341523, doi: 10.1016/j.scitotenv.2019.134153”
Response 3: Dear reviewer, thank you for your detailed opinions. Here we want to express the impact of topography on diversity in large-scale forests. We have modified it to:(L59-62) However, similar studies have mostly focused on the plant diversity of grasslands [50-52], but little is known about how topographic factors affect plant richness and whether other drivers affect hump-type diversity of mountain forest on a regional scale.
Point 4: “In addition, there is no information on vegetation type, for which the effects of environmental factors on plant species diversity are well recognized by previous authors.”
Response 4: Dear reviewer, thank you for your detailed opinions. There is some relative information on vegetation type at L65-74. In this paragraph, we emphasize the importance of site conditions, including forest types (the kind of taxa they formed), functional groups and structures.
Point 5: “L47-48 Here I would expect the opposite tendency, i.e. if temperature is higher (e.g. in tropical rain forest), then species richness and diversity is expected to be higher than in cooler regions (e.g. boreal taiga forests).”
Response 5: Dear reviewer, thank you for your detailed opinions. We agree with you on this point, but what we want to express here is not on such a large span (tropical rain forest vs boreal taiga forests). We agree that the overall trend of the latitude effect is correct, but we believe that it may be influenced by local factors. This is reflected in our results (Fig 5a, 5g). This is also the main reason why we choose the boosted regression trees model, because it can fit the nonlinear relationship between diversity and factors. We have modified it to:(L59-L62) e.g., a greater plant diversity has been observed at cooler sites compared to warmer sites in near region.
Point 6: “L50-52 Please provide more details on how insolation can affect primary production.”
Response 6: Dear reviewer, thank you for your detailed opinions. We have modified it to: (L52-53)The influence of sunshine duration and sunlight intensity on primary productivity and productivity on diversity have also been widely recognized [31–36](Charbonnier et al., 2017; Zhu et al., 2017; Jiao et al., 2018), whereas the relative importance of sunshine duration on diversity remains unclear.
Point 7: “L63 Unclear what means “given the shading effect of canopy”.”
Response 7: Dear reviewer, thank you for your kind reminding. We have modified it to: (L65-66) Plants in different functional groups (tree, shrub, herb, and treelet) have different environmental requirements when considering the shading effect of canopy [52,53].
Point 8: “L81-82 Here the authors provided one sentence on the importance of their study for forest management and conservation, but this aspect should be discussed here in concrete, i.e. how the knowledge on abiotic drivers shaping compositional patterns of forest vegetation would contribute to better understanding ecological mechanisms responsible for species assembling and why these results can be important for forestry and nature conservation.”
Response 8: Dear reviewer, thank you for your detailed opinions. We have modified it to: (L87-88) This generated information can provide support for sustainable forest management and plant diversity conservation at regional scales [6], such as which forest types or community structures can be constructed to maintain optimal biodiversity in different environmental context.
Materials and Methods
Point 9: “L85-91 Please provide more details on forest vegetation in the study site. Is this the mountain monsoon forest of temperate climate zone?”
Response 9: Dear reviewer, thank you for your detailed opinions. Yes, it is the mountain monsoon forest of temperate climate zone. We have modified it to: (L92-99) The main standing trees are deciduous broadleaved trees (Quercus mongolica, Tilia amurensis, Acer mono and Betula platyphylla et al.), deciduous coniferous trees (Larix gmelinii and Larix olgensis), evergreen coniferous trees (Pinus koraiensis, Abies nephrolepis). The main shrubs are Rhododendron aureum, Lonicera japonica, Lespedeza bicolor, Euonymus alatus et al. The main herbs are Arisaema heterophyllum, Saussurea japonica, Brachybotrys paridiformis, Oxalis corniculate et al. The regional climate is mountain monsoon forest of temperate continental climate zone and characterized by four distinctive seasons influenced by temperate monsoons, with a warm-rainy summer and a cold-dry winter.
Point 10: “L106-117 Add units in which predictors you used were measured.”
Response 10: Dear reviewer, thank you for your advice. We have modified it to: (L118-121) The annual mean temperature (AMT, °C), annual precipitation (AP, mm), and monthly (July–August) mean sunshine duration (SSD, hours) data were derived from the China Meteorological Administration website [73] for our plots. The topographical variables (elevation, m; slope, °) were obtained during the data investigation.
Point 11: “L110-111 Not clear why SD of DBH rather than simply DBH was used to characterize tree stand structure.”
Response 11: Dear reviewer, thank you for your detailed opinions. Forest age is the most appropriate indicator here, but we don't have the data. Considering the different growth rates of different tree species and the different growth rates of the same tree species in different environments, we wanted to explore the effect of tree species size differences on diversity. DBH can reflect the size of tree species, and the variance of DBH can represent the size difference of tree species in sample points.
Point 12.1: “L119-150 There is unclear which analysis was used for revealing of which pattern. Focusing on clustering and Fig. 2, one may suppose that this analysis was employed for determination of different forest types, but something different is mentioned in Methods (L. 112-113; important value of tress). Did the authors used IVs as input data for clustering performance? Please specify.”
Point 12.2: “Similar issue is linked with GBM. In two paragraphs (L. 126-143) the authors provided detailed description of GBM and its advantages in ecological modeling. However, I missed information on the linkage between employment of GBM and data used in the study.”
Point 12.3: “Moreover, focusing on Fig. 4 and description in L. 185-188, it suggests that the authors used an ANOVA to describe differences between mean richness and diversity of different functional groups of plants across different types of forest communities, but there is no information on this in methods. Correspondingly, the same concern is connected with a post-hoc analysis used to use between which forest type there were significant differences.”
Dear reviewer, thank you for your detailed opinions.
Response 12.1: We have modified it to: (L132) We used IV of trees as input data for cluster analysis.
Response 12.2: We have modified it to: (L136-137) The species richness and the factors of the 3160 plots were used in the GBM. (L136) Moreover, it can fit the nonlinear relationship continuously between diversity and factors.
Response 12.3: Here, we want to verify the result of cluster analysis and prove that the richness of each component is different, so the model analysis will be used in the following analysis. We have modified it to: (L198-199) We also compared the differences of plant richness for each component based on results of IV clustering.
Results
Point13: “L161. Table 1 is not self-explaining. Please provide what mean abbreviations used here. I suppose that abbreviations in fourth column are forest types, but what means FT?”
Response13: Dear reviewer, thank you for your detailed opinions. We have modified it to: (L175) FT is the codes of forest type and will be used in GBM.
Point14: “L175. Similar as in case of Table 1. Figure should be self-explanatory, in order to have better orientation in the text. To achieve this, maybe useful it would be to provide at least provide that for full names of forest vegetation type see Figure 2?”
Response14: Dear reviewer, thank you for your detailed opinions. We have modified it to: (L188-190) (a) Larix gmelinii forest, (b) Larix olgensis forest, (c) Mixed broadleaved forest, (d) Populus davidiana and Betula platyphylla forest, (e) Pinus koraiensis broadleaved forest, and (f) Pinus tabulaeformis broadleaved forest.
Point15: “L183. In results there is no reference to Fig. 4 (the same with Fig. 6). Please specify that boxplots with the same letters indicate not significant differences between vegetation parameters and forest type, according to the multiple comparative analysis results.”
Response15: Dear reviewer, thank you for your detailed opinions. We have modified it to: (L198-199) The same letters indicate not significant differences between forest types.
Point16: “L228. In order to make better orientation in the text, maybe you could add the percentage of average contribution of each predictor in explaining the variability of response variables?”
Response16: Dear reviewer, thank you for your detailed opinions. We have modified it in Table 2
Discussion
Point17.1: “In the whole section there is a great emphasis on the role of abiotic factors driving richness and diversity of forest communities. However, no information on impacts of biotic factors on shaping species assembling is mentioned here, that can be equally important factors in explaining differences in species composition across different forest types. Thus, it would be grateful to see how niche processes (competition and niche partitioning) may shape species richness and diversity. I know that this was not the aim of this study, but one additional paragraph, where the authors would discuss potential interactions between both abiotic and biotic factors, and how this complex of environmental drivers may result in shift in plant species composition may be useful in order to more detailed discussion of the patterns revealed (see “integrated community concept”; Lortie, C. J., Brooker, R. W., Coler, P., Kikvidze, Z., Michalet R., Pugnaire F. I., & Callaway, R. M. 2004). Rethinking plant community theory. Oikos 107(2), 433–438).
Point17.2: In this light, my another suggestion is to add some explanations (maybe in introduction), why the authors focused only on abiotic factors and completely excluded biotic drivers. Was it because previous studies found habitat filters as the most important drivers of forest composition? Also, ecological meaning of each predictor found to be significantly influencing forest diversity should be provided here in details.”
Dear reviewer, thank you for your detailed opinions.
Response17.1: We add one paragraph at L296-301 (Strong direct and indirect effects are found among plant species [104-106], and the effects between organisms is not only a response to the abiotic factors, but also a self-adaptation [107]. These interactions demonstrate that competition could produce positive and negative effects on individuals within community [108-109]. Therefore, we could not ignore the biotic factors in the ecological process, although the relative contribution of climate is higher than that of biological factors (stand structure and forest type) on large scale.)
Response17.2: we did not completely exclude biotic drivers. We believe that community structure and forest types, as biological factors, can affect plant diversity, although the contribution is small on a large scale. We introduced these two aspects at L68-72.
Point18: “L245-246 But this contradicts the statement provided in introduction (L. 47-49).”
Response18: Dear reviewer, thank you for your detailed opinions. We agree that the overall trend of the latitude effect is correct, but we believe that it may also be influenced by local factors (May be the effect is small). This is reflected in our results (Fig 5a, 5g) and we tried to analyse it (L273-285).
Point19: “L273 influencing the diversity?”
Response19: Dear reviewer, thank you for your detailed opinions. It is ‘increasing the diversity’, not influencing the diversity. Here, we want to express that complex community structures, with individuals of different ages and sizes, can promote the diversity.
Conclusions
Point20: “Some information on applicability of the patterns revealed in practice (e.g. forest management and nature conservation) may be put here or as the last paragraph of the discussion (see my comments from introduction L. 81-82).”
Response20: Dear reviewer, thank you for your detailed opinions. We add some practice about forest management at L325-330.
Reviewer 3 Report
Dear Authors,
I have reviewed the paper "Temperature-dominated driving mechanisms of the plant diversity in temperate forests, Northeast China". The aims of the paper are germane with Forests topic, in this form of article fits with the international scientific standards. The paper is written with an appropriate English level. The contribution of this paper to the scientific knowledge is average. In my opinion, there are few, but important flaws, and I suggest to improve the paper following the suggestions showed in the comments below and in the file attached:
I found none variable referred to forest management or treatment. These are parameters that can strongly influence the biodiversity of a forest stands (positively or negatively), so, if the investigated stands are actively managed forests, I think that the analysis of this parameter is needed.... The results of the study are interesting and the authors made a great effort to collect many data and this is worthy... However, in my opinion, to not link plant biodiversity in a managed forest to the forest management is an important lack, for example authors should have taken into consideration the kinds of forestry interventions, the mechanization level of forest utilization, the time laps among various interventions, etc... If the investigated stands were not managed ones I congratulate the authors for the interesting paper and I suggest only to specify the fact that these are not managed forests... on the other hand if these are active managed stands I suggest to improve the analysis taking into consideration also the mentioned above parameters or however other ones linked with forestry interventions....
You write in detail specific research objectives, but in the conclusions, they aren’t adequately developed. I suggest to add them in the conclusions as point by point findings.

Author Response
Response to Reviewer 3 Comments
Point 1: “Considering the reference format of the Journal I suggest to put these numbers linked with the hypothesis of the authors in Roman numbers, i.e. I, II, III, IV, V and VI....” (In Introduction, L76-L80).
Response 1: Dear reviewer, thank you for your advice. We have modified it to: (L80-85) We hypothesized that (I) plant diversity increases with annual mean temperature; (II) annual precipitation promotes diversity, but has an upper limit; (III) plant diversity varies with sunshine duration in hump-shaped; (IV) the greater the difference in the ontogenetic stage, the higher the plant diversity; (V) plant diversity changes with forest type; and (VI) the relative contribution of the drivers of plant diversity and the responses vary with functional groups on a regional scale.
Point 2: “I found none variable referred to forest management or treatment. These are parameters that can strongly influence the biodiversity of a forest stands (positively or negatively), so, if the investigated stands are actively managed forests, I think that the analysis of this parameter is needed....” (In Materials and Methods, 2.3. Explanatory Variables).
Point 4: “The results of the study are interesting and the authors made a great effort to collect many data and this is worthy... However, in my opinion, to not link plant biodiversity in a managed forest to the forest management is an important lack, for example authors should have taken into consideration the kinds of forestry interventions, the mechanization level of forest utilization, the time laps among various interventions, etc... If the investigated stands were not managed ones I congratulate the authors for the interesting paper and I suggest only to specify the fact that these are not managed forests... on the other hand if these are active managed stands I suggest to improve the analysis taking into consideration also the mentioned above parameters or however other ones linked with forestry interventions....”
Response 2 and 4: Dear reviewer, thank you for your advice. Our investigated stands were unmanaged and did not take any treatment during the survey. We will consider your suggestions in future studies.
Point 3: “How is it possible that a Pinus spp forests is a broadleaved stand?... I know that this is only a misunderstanding but could be confusing to the readers... I suggest, if I understood correctly, something like "Pinus spp stands with broadleaves" or "Mixed Pinus spp-broadleaves forest"” (In Results, 3.1. Forest Type Cluster, L161-L162).
Response 3: Dear reviewer, thank you for your advice. Your understanding is correctly. We have modified it to: (L168-169) mixed Pinus koraiensis-broadleaved forest (PKF), and mixed Pinus tabulaeformis-broadleaved forest (PTF).